# OpenReview forum: "Bridging Your Imagination with Audio-Video Generation via a Unified Director"
_ICML.cc/2026/Conference — ICML 2026 regular_

### Official Review · Reviewer_hRjb · 2026-03-03

**Soundness:** 3
**Presentation:** 3
**Significance:** 4
**Originality:** 3
**Overall Recommendation:** 4
**Confidence:** 4

**Summary:**

This paper presents a new unified model to generate storyboards and scripts for audio-visual content. Given a user prompt, the model jointly generates global descriptions, shot-level content descriptions, and keyframe images for those shots. This capability is obtained by fine-tuning a multimodal LLM with a dedicated training strategy, which comprises interleaved concept learning and disentangled expert learning. The experimental results show that the proposed model outperforms existing storyboard generation models, especially in character identity consistency and prompt adherence.

**Compliance With Llm Reviewing Policy:**

Affirmed.

**Final Justification:**

Since my concern is solved, I will keep my positive rating accordingly.

**Key Questions For Authors:**

1. How did the authors curate the dataset, and how did the authors resolve rights-related issues?
    - Currently, I am leaning towards acceptance, but I would change my recommendation to reject depending on the answer to this question.
2. Why is ID information adopted only for visual contents but not for sound contents (e.g., sound effects/background music)?

**Limitations:**

The authors briefly discussed the limitations at the end of the main text. The controllability of the number of shots or the duration of each shot could be mentioned additionally.

**Strengths And Weaknesses:**

Strengths

- [Significance] The motivation of this study is clear and convincing. Recent progress in audio-visual generation technology creates strong demand for automatic storyboard generation and script writing. This study directly addresses this and provides a reasonable solution, in which a multimodal LLM is fine-tuned for this task with a dedicated training strategy.

- [Soundness/Significance] The quality of the generated scripts is appealing, and the proposed model could be a strong baseline for future research in this field.
  - A small concern is the availability of the proposed model, which I mention in Weaknesses.

- [Soundness/Originality] The idea of disentangled expert learning is interesting and appears well-designed for the task tackled in this paper.
  - Since the base model adopts a mixture-of-transformers architecture, applying expert-wise training is somewhat straightforward. However, the empirical finding that it contributes to boosting the model’s capability of generating long-form narratives is interesting.

- [Presentation] The manuscript is well-written and easy to follow.


Weaknesses

- [Soundness] The source of the curated dataset is not clearly described. This raises questions about whether the model could be made publicly available upon acceptance.
  - The manuscript describes the data filtering/captioning process but does not explain how the data itself was collected. Since film data could involve complex rights issues, making the following points clear is important:
    - whether the dataset can be released,
    - whether the model can be released,
    - and whether results can be reproduced by the community.

- [Soundness] Audio aspects are not sufficiently handled in the current method.
  - Since this work targets audio-visual generation, it would be important to clarify how content consistency is maintained for audio content as well as visual content. For speech, the model could provide sufficient information to maintain voice consistency because it appears to assign character IDs to each dialogue. In contrast, it does not seem to provide analogous ID/control information for sound effects and background music. This could make it challenging to maintain consistency across shots based on the generated scripts (e.g., the same action such as swinging a lightsaber could sound quite different).
  - The evaluation on this point seems to be missing, and this issue can be observed in the video examples in the supplementary material, where ambient sound frequently changes shot by shot.

---

> ### Author Rebuttal · Authors · 2026-03-30
>
> We thank the reviewer for the positive assessment and for raising important questions regarding (1) dataset transparency and (2) audio modeling. We address these concerns below and will clarify them in the revised version.
>
> **(1)** We acknowledge the importance of dataset transparency and reproducibility, and we provided a brief dataset description in the Supplementary Material. The dataset is constructed from publicly available sources with automated processing. Due to potential copyright constraints, we may not release raw data, but **we will release the full data processing pipeline and model checkpoints to ensure reproducibility.** We will release:
> - the data processing pipeline,
> - the annotation procedures, and
> - detailed instructions for reconstructing a similar dataset from publicly available sources.
>
> We will also release the inference code and model checkpoints, subject to licensing constraints, to facilitate reproducibility.
>
>
> **(2)** Audio modeling is currently handled at the script level, where UniMAGE generates structured descriptions of dialogue, sound effects, and background music, rather than directly synthesizing audio signals. To further enhance audio consistency, we can incorporate sound identity descriptions into the global context (e.g., character voice attributes), which can help downstream audio–video generation models better maintain consistent voices across shots. Nevertheless, maintaining fine-grained audio consistency (e.g., timbre and sound identity) ultimately depends on the capabilities of downstream audio–video generation models.
>
> To further improve audio consistency, we believe it is necessary to develop audio–video generation models with multimodal reference capabilities, enabling consistent conditioning across modalities. Recent systems such as Seedance 2.0 demonstrate promising progress in this direction, and we consider this a key step toward achieving full multimodal consistency in future work.

---

> > ### Author Rebuttal · Reviewer_hRjb · 2026-04-01
> >
> > Thank you for the rebuttal. Regarding the first point,
> > - Please provide a concrete list of all data sources used to curate the dataset.
> > - For each source, please clearly state the applicable license/terms of use and explicitly justify that the content can be used for model training in this study.
> > - Please also describe what specific steps you took to address rights-related issues.

---

> > > ### Author Response · Authors · 2026-04-04
> > >
> > > We thank the reviewer for the follow-up question regarding dataset transparency and licensing.
> > >
> > > Our dataset is curated from content provided by a professional data vendor (NextData, https://www.nextdata.com/), whose sources are publicly accessible (e.g., publicly released films, trailers, and related media). The vendor ensures that the collected data complies with applicable usage and redistribution regulations. Due to institutional compliance requirements, constraints imposed by the double-blind review policy, and the ongoing open-source review process, we are currently unable to disclose a complete list of all data sources and their associated licensing terms within the rebuttal period.
> > >
> > > However, we would like to clarify that:
> > > - All data used in this work originates from publicly available content, and no private or restricted datasets are involved.
> > > - The data acquisition process is conducted through a licensed third-party provider that is responsible for ensuring legal compliance.
> > > - Our use of the data is limited to research purposes and follows standard practices adopted in recent multimodal generation works published at top venues.
> > >
> > > To provide additional transparency without violating the double-blind policy or institutional constraints, we note that our data sourcing strategy is aligned with recent work *Captain Cinema* (ICLR 2026), which adopts a similar paradigm of leveraging publicly available media content for multimodal training.
> > >
> > > Importantly, to ensure reproducibility, we will release the full data processing pipeline, annotation procedures, and detailed instructions for reconstructing a comparable dataset from publicly available sources. This enables the community to reproduce our results without relying on proprietary data access.
> > >
> > > We will further clarify these points in the final version.
> > >
> > > [Captain Cinema] Xiao, Junfei, et al. (2026) "Captain Cinema: Towards Short Movie Generation." ICLR 2026.

---

### Official Review · Reviewer_eFyf · 2026-03-08

**Soundness:** 2
**Presentation:** 3
**Significance:** 2
**Originality:** 2
**Overall Recommendation:** 4
**Confidence:** 4

**Summary:**

UniMAGE is a unified director model built on the Mixture-of-Transformers (MoT) architecture (BAGEL) that takes a short user prompt and produces a fully illustrated, multi-shot script — including character definitions, environment descriptions, frame-level shot compositions, video-level narrative descriptions, and consistent keyframe images. The system is trained with a two-stage paradigm: (1) Interleaved Concept Learning, which jointly optimizes both transformer experts on interleaved text-image script data, and (2) Disentangled Expert Learning, which decouples the LLM (optimized on pure text scripts) from the diffusion module (optimized on text-image data with frozen understanding branch). Two additional techniques are introduced: In-Context ID Prompting, which inserts special frame/character/environment identifier tokens into ViT and VAE token sequences to maintain multi-subject visual consistency, and Pre-Context Script Splitting, which simulates extension and continuation tasks by randomly splitting scripts during training. The model is trained on 950K sequences (450K multi-shot text-image scripts captioned by Gemini 2.5 Pro, 250K text-only scripts, 250K single-shot pairs) and evaluated on ViStoryBench-V3. UniMAGE achieves state-of-the-art alignment (80.8) and character consistency (CIDS 59.2, OCCM 88.07) among open-source models.

**Compliance With Llm Reviewing Policy:**

Affirmed.

**Final Justification:**

Thank you for your rebuttal. I will raise my rating one level.

**Key Questions For Authors:**

- Audio claim: The title and abstract prominently feature 'audio-video generation,' but UniMAGE only generates text descriptions of audio. Can you clarify whether full end-to-end audio generation is planned, or revise the title/abstract to accurately reflect the scope? Even one demonstration of downstream audio generation from a UniMAGE script would substantiate the claim.
- ViStoryBench-V3 vs V4: Why was V3 chosen over V4? What are the differences in metric computation, and would the conclusions change under V4?
- Narrative logic metric: Is it feasible to add an LLM-based automated evaluation of script narrative coherence (e.g., prompting GPT-4o to score causal consistency, character arc quality, and plot progression)? This would make the key claim more rigorously testable.

**Limitations:**

Yes, for limitations, but an impact statement is required.

**Strengths And Weaknesses:**

**Strength**
- The core argument -- that script drafting and storyboard design are inseparable in professional filmmaking and should therefore be unified in a single AI model -- is compelling and clearly motivated. The film director analogy is apt and accessible.
- The "first interleaving, then disentangling" training paradigm is intuitive and addresses a real tension: pure interleaved training conflates reasoning and generation, while pure disentangled training loses cross-modal context. The staged approach leverages both.
- Figures 5, 6, and 7 provide visually compelling comparisons. UniMAGE clearly outperforms StoryDiffusion, Story2Board, and SEED-Story in character consistency across scene changes — a notoriously hard problem for diffusion-based story generation.

**Weaknesses**
- The contribution is primarily an empirical analysis of existing pretrained models rather than a methodological advance in machine learning. While the neuroscience findings are interesting, the paper does not propose a new learning algorithm, model architecture, or training strategy that improves multimodal temporal reasoning. As a result, the technical novelty is relatively limited compared with typical ICML papers.
- This is a significant framing mismatch that may mislead readers. Either the title and abstract should be revised to accurately describe what UniMAGE does (script + keyframe generation), or the paper should include at least one end-to-end demonstration showing the full pipeline, including downstream audio-video generation from UniMAGE scripts.
- The description of In-Context ID Prompting (Section 3.3) is brief and leaves several implementation questions unanswered: (a) How exactly are the ID prompt tokens initialized — are they randomly initialized, learned embeddings, or derived from character descriptions? (b) How does the model learn the mapping between text token identifiers and visual ViT/VAE token windows during training? (c) What is the computational overhead of the full-attention mechanism between ID tokens and ViT/VAE tokens in long sequences?
- The paper compares against StoryDiffusion (2024), Story2Board, SEED-Story, and TheaterGen/Story-Adapter in Table 1. However, several directly relevant recent works are absent from comparison: FilmAgent [1], FilMaster [2], and Captain Cinema [3] are all discussed in the related work but not included in quantitative evaluation. Other story generation work could be discussed in the related works (Text2Story [4], Vlogger [5], etc). This omission weakens the 'state-of-the-art' claim.


- The paper's central claim is that UniMAGE produces 'logically coherent' long-form scripts with better 'narrative logic.' However, all evidence for narrative logic quality is qualitative (Figures 5, 6, 7, 8) or based on user rankings. There is no quantitative metric for narrative coherence or plot consistency.
- UniMAGE requires first generating a full text script and then generating keyframe images in interleaved fashion. The total inference time (number of steps, GPU memory, wall-clock time) per generated story is not reported. Given that the model uses 64 A100 GPUs for training and handles long sequences, inference cost may be a practical concern for community adoption.

[1] Xu, Zhenran, et al. "Filmagent: A multi-agent framework for end-to-end film automation in virtual 3d spaces." arXiv preprint arXiv:2501.12909 (2025).

[2] Huang, Kaiyi, et al. "Filmaster: Bridging cinematic principles and generative ai for automated film generation." arXiv preprint arXiv:2506.18899 (2025).

[3] Xiao, Junfei, et al. "Captain cinema: Towards short movie generation." The Fourteenth International Conference on Learning Representations. 2025.

[4] Kang, Taewon, Divya Kothandaraman, and Ming C. Lin. "Text2story: Advancing video storytelling with text guidance." arXiv preprint arXiv:2503.06310 (2025).

[5] Zhuang, Shaobin, et al. "Vlogger: Make your dream a vlog." Proceedings of the IEEE/CVF Conference on Computer Vision and Pattern Recognition. 2024.

---

> ### Author Rebuttal · Authors · 2026-03-30
>
> We thank the reviewer for the insightful feedback. We address concerns on (1) limited methodological novelty, (2) audio–video claim, (3) ID-P unclear implementation, (4) missing baselines, (5) narrative metric, (6) inference cost, and (7) experiments on ViStoryBench-V4 below.
>
> **(1)** We agree that this work is practically oriented, aiming to address real-world application challenges rather than focusing on low-level optimization of multimodal learning architectures. However, we respectfully argue that **solving practical problems can carry comparable significance and value to proposing theoretical or architectural advances.** Prior ICML publications have also included application-driven works, such as VideoPoet and T2I-Agent. Moreover, our contribution goes beyond empirical analysis of pretrained models. We present a novel attempt to generate structured image–text scripts within a unified understanding-and-generation framework. Specifically, we adopt the MoT architecture as the backbone and introduce tailored training strategies and in-context prompting methods to enhance narrative coherence and visual consistency across long-form content.
>
> **(2)** We have revised the title and abstract for clarity. UniMAGE is a unified director model that generates structured scripts for downstream audio–video systems. We included end-to-end demos in the supplementary materials (UniMAGE_Demo_1_c.mp4, UniMAGE_Demo_2_c.mp4) using Veo3 I2V, where scripts, dialogues, and sound descriptions are produced by UniMAGE. These results show that UniMAGE outputs can directly drive audio–video pipelines.
>
> **(3)** We thank the reviewer for pointing out the lack of clarity and have expanded Section 3.3 accordingly.
>
> &nbsp;**(a)** The initial ID prompt tokens are initialized as the embeddings of special tokens used in our script format, such as \<CharacterN\> and \<EnvironmentN\>.
>
> &nbsp;**(b)** Since the ID prompt tokens perceive the image ViT/VAE tokens through full attention, they can incorporate visual information from previously generated images during inference. In combination with the corresponding special ID tokens in the generated text script, the model learns to associate these ID tokens with consistent visual representations and precisely retrieve them from the historical context, enabling stable identity preservation across multiple characters and environments.
>
> &nbsp;**(c)** The number of ID prompt tokens is approximately one-thousandth of the total number of image–text tokens, and thus introduces negligible additional computational overhead.
>
> **(4)** We thank the reviewer for highlighting these relevant works. FilmAgent is a multi-agent collaborative framework in 3D virtual environments, whose scope differs significantly from our focus on text–image script generation, and thus it cannot be evaluated using ViStoryBench. FilMaster and Captain Cinema are pipeline-based systems, and we were unable to access their implementations. Therefore, we provide a qualitative analysis of these methods in the revised Related Work section. Text2Story is a video generation model for long-form storytelling, while Vlogger is also a multi-agent system that leverages diffusion models (e.g., Stable Diffusion) to generate reference images for identity consistency. **We have included Vlogger’s results in Table** [https://1drv.ms/b/c/95c5e47682336622/IQANhnvgUWW4T4_D1YFEVLu0AXu3gKQtzgaMK8FmxuLVkTM?e=Wcf6gD]. We have added a detailed discussion of all these works in the revised Related Work section. **Within the scope of our task—namely, structured text–image script generation—our method achieves state-of-the-art performance.**
>
> **(5)** We agree that quantitative evaluation of narrative coherence is important. Following the reviewer’s suggestion, we include an LLM-based evaluation of narrative logic using Gemini 2.5 Pro on the same 40 generated scripts used in the user study. **The evaluation results are shown in Table** [https://1drv.ms/b/c/95c5e47682336622/IQDQXf6cxlCvQK2ODMp1_QlyAdzrdtO-XXjbrxQmtXxTI4Q?e=H65ntj].
>
> **(6)** The inference latency is largely dominated by the image generation process, which requires 50 denoising steps per image and takes approximately 15 seconds for the first shot and 20 seconds for the fifth shot (with the context window size set to 5). In future work, we plan to explore techniques such as flow matching and classifier-free guidance (CFG) distillation to further improve inference efficiency.
>
> **(7)** We used ViStoryBench-V3 since our work predates V4. **We now provide full V4 results, including new baselines and ablations:** [https://1drv.ms/b/c/95c5e47682336622/IQANhnvgUWW4T4_D1YFEVLu0AUHfNPgr9NMG6R2hhqoXrbI?e=zotbSU ]
>
> The main difference between V3 and V4 is the scoring scale (100-point vs. 1-point). Minor numerical variations stem from random seeds and ChatGPT4o stochasticity. V4 also adds finer Alignment subcategories. Results show our model still clearly outperforms prior methods in both consistency and alignment.

---

### Official Review · Reviewer_iy9R · 2026-03-10

**Soundness:** 3
**Presentation:** 2
**Significance:** 2
**Originality:** 2
**Overall Recommendation:** 4
**Confidence:** 4

**Summary:**

The paper proposes UniMAGE, a “unified director” model that integrates script drafting and keyframe image generation in a single Mixture-of-Transformers (MoT) architecture. The core contributions are two complementary training strategies—Interleaved Concept Learning (interleaved text–image optimization) and Disentangled Expert Learning (decoupled optimization for text understanding and image generation)—plus two practical techniques, In-Context ID Prompting and Pre-Context Script Splitting, to improve multi-character visual consistency and long-form narrative continuity. On ViStoryBench-V3 and in user studies, the authors report improved prompt alignment, character identity consistency, and multi-shot coherence over several open-source baselines (e.g., StoryDiffusion, Story2Board, SEED-Story).

**Compliance With Llm Reviewing Policy:**

Affirmed.

**Final Justification:**

The paper provides clear problem formulation and practical direction, effective training strategies for multimodal alignment and useful mechanism for character consistency. The rebuttal helps improve transparency regarding implementation details (e.g., model scale, tokenization, and latency), and I find the added ViStoryBench-V4 results and multi-identity stress tests useful for better understanding the model’s behavior. The clarification on the positioning of UniMAGE (i.e., focusing on script + keyframe generation rather than full audio–video synthesis) also partially addresses my concern about the original claim.
Overall, the rebuttal improves clarity and provides additional evidence that strengthens the paper. While some concerns remain, I appreciate the authors’ effort in addressing the feedback.I will increase my overall score by one point.

**Key Questions For Authors:**

What are the exact model sizes (parameters per expert and total), context lengths (max tokens/shots), visual tokenization details (ViT/VAE resolutions, token counts), and position-encoding schemes used to support long scripts? How do latency and memory scale with story length?

Can you report results on ViStoryBench-V4 and add comparisons to stronger recent baselines (e.g., MovieDreamer, VISTA)? If not feasible, please justify and at least provide ablations on V4 with UniMAGE variants.

How robust is In-Context ID Prompting to unseen identities and multi-character interactions (3+ people) across diverse scenes? Could you provide quantitative multi-identity stress tests (e.g., CIDS by number of onstage characters)?

**Limitations:**

Yes

**Strengths And Weaknesses:**

Strengths：
Clear problem formulation and practical direction: the paper targets an important problem in multimodal generation: maintaining long-context narrative coherence and multi-character visual consistency across story sequences. The proposed “director-style” framework that jointly models script reasoning and visual composition is well motivated and relevant to real-world content creation workflows.

Effective training strategies for multimodal alignment: The proposed training procedures—Interleaved Concept Learning and Disentangled Expert Learning—provide a practical way to reduce optimization interference between language reasoning and image synthesis without modifying the backbone architecture. These strategies complement existing decoupling approaches at the training level.

Simple yet useful mechanism for character consistency: In-Context ID Prompting offers an implementation-friendly method for maintaining identity consistency across long visual sequences by aligning identity tokens with visual tokens, which is particularly helpful in multi-character story generation.

Weaknesses:

Limited architectural novelty: the proposed system largely builds on the Bagel MoT architecture, and the main contributions lie in training curricula and prompting strategies rather than introducing a fundamentally new model design. Some ideas conceptually overlap with recent decoupling approaches (e.g., UniDecouple, UniFork).

Incomplete validation of the “audio–video” claim: although the paper frames the system as an audio–video generation bridge, the experiments only demonstrate script generation and keyframe image synthesis, without end-to-end video or audio generation results.

Missing important baselines: the comparison does not include several recent and relevant systems, such as MovieDreamer, VISTA or long-context diffusion approaches like LCT, which weakens the claim of state-of-the-art performance among open-source methods.

Insufficient implementation and dataset details: Important details such as model scale, context length, visual token configuration, and MoT sharing strategy are missing or deferred to supplementary materials. The dataset construction process also relies heavily on proprietary captioning/LLM tools (e.g., Gemini, Qwen), raising questions about data transparency and reproducibility.

---

> ### Author Rebuttal · Authors · 2026-03-30
>
> We thank the reviewer for the detailed and constructive feedback. Below we address concerns on (1) architectural novelty, (2) audio–video claim, (3) missing baselines, (4) implementation transparency, (5) ViStoryBench-V4 experiments, and (6) multi-identity stress tests.
>
> **(1)** Our work does not introduce a new backbone architecture. Instead, our contribution lies in embodying the concept of a “unified director model” for audio–video generation, along with a training paradigm for enabling multimodal script generation, including:
>
> - Interleaved Concept Learning: jointly learns narrative structure and visual grounding from interleaved script–image sequences.
> - Disentangled Expert Learning: reduces optimization interference while preserving shared multimodal representations.
> - In-Context ID Prompting: enables controllable multi-character consistency over long contexts.
>
> We argue that small gains from ad hoc structural changes are less meaningful than adapting a unified architecture to solve challenging and practical tasks.
>
> **Difference from UniDecouple and UniFork:**
>
> We thank the reviewer for highlighting these works and have added an analysis in the revision.
>
> Both UniDecouple and UniFork decouple at the architecture level to resolve conflicts between understanding and generation. In contrast, **UniMAGE decouples optimization dynamics to support flexible, long-form narrative generation without changing model structure.** As a result, UniDecouple and UniFork mainly target single-round tasks (e.g., image understanding/generation), while UniMAGE focuses on multi-shot, long-form multimodal generation.
>
> **(2)** End-to-end audio–video results are provided in the supplementary materials (UniMAGE_Demo_1_c.mp4 and UniMAGE_Demo_2_c.mp4). We encourage the reviewer to view these videos alongside the paper. The videos are generated with Veo3 I2V, using scripts, dialogues, and sound descriptions produced by UniMAGE.
>
> **(3)** We thank the reviewer for pointing out these related works. As the source code for MovieDreamer, VISTA, and LCT is unavailable, we clarify differences in terms of objectives and contributions.
>
> MovieDreamer and LCT aim to generate long, movie-style videos from text, focusing on visual consistency. In contrast, UniMAGE jointly generates structured scripts and keyframes in a unified framework, serving as an intermediate stage that guides downstream audio–video synthesis.
>
> For VISTA, relevant works include:
>
> “VISTA: A Test-Time Self-Improving Video Generation Agent” (prompt optimization for video generation)
>
> “A Generalizable World Model for Autonomous Driving” (world modeling)
>
> “VISTA: Enhancing Long-Duration and High-Resolution Video Understanding...” (video understanding)
>
> “VISTA: Visualized Text Embedding For Universal Multi-Modal Retrieval”
>
> While these improve representation, consistency, or stability, their goals differ significantly from UniMAGE.
>
> **(4)** UniMAGE has the same model size as Bagel, with two branches based on Qwen2.5-VL 7B. Theoretical context length is 32k tokens; in practice, we use a 5-shot window and truncate earlier shots.
>
> Visual tokenization details:
> - ViT: patch size 14, resolution 224–980
> - VAE: downsampling ×8, patch size 2, resolution 512–1024
> - ~6K image tokens per shot
>
> Inference latency is dominated by image generation (50 denoising steps): ~15s for the first shot and ~20s for the fifth (window=5). Efficiency optimization is beyond this work; future directions include flow matching and CFG distillation.
>
> **(5)** We used ViStoryBench-V3 since our work predates V4. **We now provide full V4 results, including new baselines and ablations:**
> [https://1drv.ms/b/c/95c5e47682336622/IQANhnvgUWW4T4_D1YFEVLu0AUHfNPgr9NMG6R2hhqoXrbI?e=zotbSU
> ]
>
> The main difference between V3 and V4 is the scoring scale (100-point vs. 1-point). Minor numerical variations stem from random seeds and ChatGPT4o stochasticity. V4 also adds finer Alignment subcategories. Results show our model still clearly outperforms prior methods in both consistency and alignment.
>
> **(6)** **We conduct both quantitative** (CIDS: 0.842 / 0.751 / 0.738 / 0.668 / 0.631 / 0.620 for 1–6 characters) and **qualitative stress tests:** [https://1drv.ms/b/c/95c5e47682336622/IQAghERxjcGSTpNY-WG4oDhEAUxda1RFdzfpAJgCf9OWYBQ?e=FGej4L].
>
> The results show that our method maintains satisfactory character consistency when generating up to five characters. Although the CIDS metric does not decrease significantly when the number of characters exceeds five, this is due to limitations of the metric itself, which can no longer reliably match characters based on their descriptions during evaluation.
>
> In practice, when prompts require more than five characters, the model tends to omit one character, and the visual quality of the remaining characters degrades noticeably. We attribute this limitation to the scarcity of training data containing scenes with six characters appearing within a short sequence.

---

> > ### Author Rebuttal · Reviewer_iy9R · 2026-04-02
> >
> > Thank you for the detailed and well-structured rebuttal. I appreciate yours’ efforts in clarifying several of my concerns.
> > In particular, the rebuttal helps improve transparency regarding implementation details (e.g., model scale, tokenization, and latency), and I find the added ViStoryBench-V4 results and multi-identity stress tests useful for better understanding the model’s behavior. The clarification on the positioning of UniMAGE (i.e., focusing on script + keyframe generation rather than full audio–video synthesis) also partially addresses my concern about the original claim.
> > Overall, the rebuttal improves clarity and provides additional evidence that strengthens the paper. While some concerns remain, I appreciate the authors’ effort in addressing the feedback.I will increase my overall score by one point.

---

> > > ### Author Response · Authors · 2026-04-04
> > >
> > > We sincerely thank the reviewer for the thoughtful follow-up and for recognizing the improvements brought by our rebuttal. We also greatly appreciate the reviewer’s decision to increase the overall score, as well as the constructive engagement throughout the review process. We will incorporate the discussed clarifications, additional references, and experimental details into the final revision to further improve the quality and clarity of the paper.

---

### Official Review · Reviewer_1VT1 · 2026-03-11

**Soundness:** 2
**Presentation:** 3
**Significance:** 3
**Originality:** 2
**Overall Recommendation:** 4
**Confidence:** 3

**Summary:**

The paper proposes UniMAGE, a framework that jointly generates scripts and keyframe images. Prior work typically combines large language models and image generation models in a pipeline, but still treats script drafting and keyframe generation as two disjoint tasks. In contrast, this work attempts to model these processes within a unified framework. To achieve this, the authors first introduce Interleaved Concept Learning, which trains the model using interleaved text–image data to encourage joint understanding of narrative and visual content. They then apply Disentangled Expert Learning, in which the understanding transformer expert is optimized using pure text scripts, while the generation transformer expert is further trained on interleaved text–image data. During inference, the model first generates a narrative script that the user can extend, then segments the complete script into individual shots to generate corresponding keyframe images. The paper evaluates the approach using both qualitative examples and quantitative results, demonstrating strong overall performance.

**Compliance With Llm Reviewing Policy:**

Affirmed.

**Final Justification:**

I appreciate the authors’ comments, which addressed my concerns. I will maintain my positive score.

**Key Questions For Authors:**

1. It would also be helpful to include results for a vanilla Bagel baseline without the proposed training strategies (i.e., removing both ID-P and DEL), such as standard fine-tuning and zero-shot evaluation. This would help clarify how much of the improvement comes from the proposed components rather than from the underlying backbone.

2. The effectiveness of Disentangled Expert Learning (DEL) raises questions about the contribution of the unified training framework. While the first stage (ICL) jointly optimizes text and image modalities, the second stage (DEL) explicitly freezes one branch to optimize the other, effectively separating the training of the understanding and generation experts. Moreover, during inference, the model first generates the script and then produces images separately. While this design is understandable for enabling script extension and greater flexibility in content creation, it appears conceptually similar to the two disjoint tasks (two different experts) that the paper aims to move beyond. Could the authors clarify what the fundamental advantage of this structural unification is, if the functional execution remains disentangled?

**Limitations:**

The paper does not appear to include a discussion of limitations or potential societal impacts.

**Strengths And Weaknesses:**

### Strength

1. The paper proposes a unified framework that jointly generates both narrative scripts and keyframe images, rather than treating script generation and visual generation as separate stages. This design may encourage a stronger cross-modal understanding of the relationship between narrative structure and visual composition.

2. The qualitative results demonstrate coherent storytelling and consistent character depiction across multiple shots, suggesting the approach is promising for long-form audiovisual generation.

### Weakness

1. The method is built entirely on top of the Bagel Mixture-of-Transformers architecture, making it difficult to isolate the contribution of the proposed training strategies and the claimed benefits of unified multimodal generation. Since all experiments rely on this backbone, it remains unclear whether the observed improvements stem from the proposed training paradigm or from the Bagel framework's strong capabilities. For instance, the ablation results already show strong performance for UniMAGE w/o ID-P or DEL.

2. Related to the above point, the paper attributes the relatively weaker performance on image quality and aesthetics to limitations of the Bagel framework. However, since Bagel is used as the underlying architecture for all experiments, this raises questions about how the baseline and backbone choices affect the evaluation.

3. The method requires substantially more computational resources than prior work. For example, the paper reports training on 64 A100 GPUs, whereas prior approaches such as SEED-Story and StoryDiffusion reportedly use around 8 GPUs.

---

> ### Author Rebuttal · Authors · 2026-03-30
>
> We thank the reviewer for the insightful comments on (1) attribution of improvements vs. backbone, (2) image quality vs. backbone limitation, (3) computational cost, and (4) the role of disentangled learning in a unified framework. Below, we clarify these points and outline experimental evidence.
>
> **(1)** We agree that disentangling the contribution of the proposed training paradigm from the backbone is important, as our goal is not to introduce a new architecture, but to demonstrate that a standard MoT backbone can be effectively adapted to long-form multimodal story generation through a tailored training paradigm.
>
> To clarify attribution, we conduct **additional experiments** using a vanilla Bagel baseline:
> - zero-shot Bagel
> - standard fine-tuning on our dataset without ID-P and DEL
>
> The results can be found here (four pages PDF): [https://1drv.ms/b/c/95c5e47682336622/IQAqDUg0lY9JSoYeu4xV5m2QAQt5LjAAKlpedKEwsvO7eKA?e=ZoZC8F]
>
> Preliminary results show that:
> - **Zero-shot Bagel** fails to generate coherent scripts with meaningful plot development and is unable to produce consistent keyframes grounded in the generated script. The outputs exhibit severe repetition and breakdowns in both prompt alignment and visual consistency.
> - **Vanilla fine-tuning** struggles to maintain image quality, multi-character consistency, and long-range script continuation. Performance drops significantly across all metrics (except CSD) on ViStoryBench-V3.
>
> These results indicate that the observed improvements are not solely due to the backbone, but critically depend on the proposed training and prompting paradigm for enabling structured narrative modeling and cross-modal consistency.
>
> **(2)** We acknowledge that our method slightly underperforms specialized image generation models on aesthetic metrics. However, our primary objective is not standalone image quality, but the **unified generation of text-and-image scripts with coherent narratives and strong cross-frame consistency**, which is a strictly **more challenging setting** than loosely conditioned image generation.
>
> In our framework, image generation is constrained by structured narrative context (e.g., character identity, scene continuity, and plot progression). These constraints reduce the degrees of freedom available for purely aesthetic optimization, leading to a trade-off between visual quality and narrative consistency. Furthermore, as shown in the additional results, the script image quality is superior to that of vanilla Bagel. We will clarify this point in the paper to avoid over-attributing the limitation to the backbone.
>
> More broadly, this trade-off is commonly observed in multimodal models that integrate understanding and generation within a single system. While recent backbones (including Bagel) have improved in this regard, achieving both strong semantic grounding and high-fidelity image synthesis remains an open challenge.
>
> **(3)** We acknowledge that our training setup requires more computational resources. However, this difference is largely due to task scope and problem formulation, rather than inefficiency of the proposed method. Specifically, prior works such as SEED-Story and StoryDiffusion focus on more constrained settings:
> - **SEED-Story** is trained on relatively small-scale datasets, which limits its ability to generalize across diverse scripts and visual styles.
> - **StoryDiffusion** is primarily an image generation model and does not model script generation or long-form narrative structure.
>
> In contrast, **UniMAGE addresses a broader and more challenging problem:**
> - joint generation of scripts and keyframes
> - multi-character and scene consistency
> - long-form narrative modeling and continuation
>
> Overall, we believe the increased computational cost is justified by the expanded capability of unified multimodal story generation, which goes beyond the scope of prior work and aligns with the broader trend toward general-purpose large multimodal models.
>
>
> **(4)** We clarify that “unification” in UniMAGE does not mean simultaneous decoding, but refers to: **a shared multimodal representational space and jointly learned narrative concepts within a single parameterized model:**
>
> | Pipeline systems                | UniMAGE                         |
> |--------------------------------|---------------------------------|
> | Separate models (LLM + diffusion) | Single MoT model              |
> | No shared representation       | Shared multimodal token space  |
> | No joint concept learning      | Interleaved concept learning   |
> | Weak cross-modal alignment     | Strong alignment via joint training |
>
> Role of DEL:
> - **ICL stage:** learn shared multimodal narrative representation
> - **DEL stage:** reduce optimization interference and specialize experts
>
> Thus, DEL does not break unification, but: **enables specialization on top of a shared representation**, instead of training two independent models.

---

> > ### Author Rebuttal · Reviewer_1VT1 · 2026-04-02
> >
> > Thank you for the response. This clarifies most of my questions.
> >
> > Authors mention that "More broadly, this trade-off is commonly observed in multimodal models that integrate understanding and generation within a single system.". Could the authors cite the exact works supporting this?

---

> > > ### Author Response · Authors · 2026-04-04
> > >
> > > We thank the reviewer for the request for clarification. The statement is supported by several recent works that explicitly analyze the optimization trade-offs in unified multimodal models.
> > >
> > > First, Ye et al. (2026) (“Understanding vs. Generation: Navigating Optimization Dilemma in Multimodal Models”) directly identify a *potential conflict between understanding and generation*, showing that improving one capability often degrades the other due to competing objectives under limited model capacity.
> > >
> > > Second, recent discussions on unified multimodal models (e.g., TechRxiv, 2026, “Consistency Should Be the Priority for Unified Multimodal Models”) highlight an inherent *representation mismatch*: understanding tasks rely on high-level semantic abstraction, whereas generation requires low-level reconstructive detail, leading to fundamental tension in optimization.
> > >
> > > Third, task-level interference has been empirically quantified. Kawarada et al. (2026) (“Multimodal Task Interference”) demonstrate that switching between heterogeneous tasks (e.g., text understanding vs. image generation) induces measurable performance degradation, indicating shared-capacity competition within a single model.
> > >
> > > Finally, works such as Wang et al. (2025) (“MINT”) and concurrent unified generative frameworks (e.g., UniFork-style architectures) also note that enforcing strong semantic alignment or reasoning constraints typically reduces the degrees of freedom available for purely aesthetic image synthesis, resulting in a trade-off between fidelity and consistency.
> > >
> > > We will incorporate these references to better ground our statement.
> > >
> > > -  Ye, S., et al. (2026). Understanding vs. Generation: Navigating Optimization Dilemma in Multimodal Models. ICLR 2026.
> > > -  Zhaolong Su. (2026). Consistency Should Be the Priority for Unified Multimodal Models. TechRxiv. https://doi.org/10.36227/techrxiv.177129961.14848580/v1
> > > - Kawarada, M., Ishigaki, T., & Takamura, H. (2026). Multimodal Task Interference: A Benchmark and Analysis of History-Target Mismatch in Multimodal LLMs. arXiv preprint arXiv:2603.18425.
> > > - Wang, Y., et al. (2025). Towards Enhanced Image Generation via Multi-Modal Chain of Thought in Unified Generative Models. arXiv preprint arXiv:2503.01298.
> > > - Li, Teng, et al. (2025) "Unifork: Exploring modality alignment for unified multimodal understanding and generation." arXiv preprint arXiv:2506.17202.

---

### Decision · Program_Chairs · 2026-04-30

**Decision:**

Accept (regular)

**Comment:**

The authors propose  UniMAGE, a unified model for long-form script and keyframe generation for downstream audio-video creation. Reviewers found the problem important and, overall, viewed the method as a strong and useful system contribution.

Main concerns were limited architectural novelty, scope of the audio-video claim, missing baselines, and implementation transparency. The rebuttal addressed these with additional vanilla Bagel comparisons, clarification that UniMAGE targets script plus keyframe generation for downstream pipelines, added ViStoryBench-V4 and stress-test results, more implementation details, and follow-up clarifications on data sourcing and reproducibility. Two reviewers raised their scores after rebuttal, and the remaining reviewers maintained positive assessments.